# Lower limb muscle fatigue after uphill walking in children with unilateral spastic cerebral palsy

I. Moll[1,2,3]*, J. M. N. Essers[2], R. G. J. Marcellis[4], R. H. J. Senden[4], Y. J. M. Janssen-Potten[5,6], R. J. Vermeulen[1,3], K. Meijer[2]

**1** School of Mental Health and Neurosciences (MHeNs), Faculty of Health, Medicine and Life Sciences (FHML), Maastricht University, Maastricht, the Netherlands, **2** Department of Nutrition and Movement Sciences, FHML, School of Nutrition and Translational Research in Metabolism (NUTRIM), Maastricht University, Maastricht, the Netherlands, **3** Department of Neurology, Maastricht University Medical Center (MUMC+), Maastricht, the Netherlands, **4** Department of Physiotherapy, MUMC+, Maastricht, the Netherlands, **5** Adelante Centre of Expertise in Rehabilitation and Audiology, Hoensbroek, the Netherlands, **6** Research School CAPHRI, Department of Rehabilitation Medicine, Maastricht University, Maastricht, the Netherlands

\* irene.moll@mumc.nl

**Data Availability Statement:** All relevant data are within the paper and its Supporting Information files.

## Abstract

Fatigue during walking is a common complaint in cerebral palsy (CP). The primary purpose of this study is to investigate muscle fatigue from surface electromyography (sEMG) measurements after a treadmill-based fatigue protocol with increasing incline and speed in children with CP with drop foot. The secondary purpose is to investigate whether changes in sagittal kinematics of hip, knee and ankle occur after fatigue. Eighteen subjects with unilateral spastic CP performed the protocol while wearing their ankle-foot orthosis and scored their fatigue on the OMNI scale of perceived exertion. The median frequency (MF) and root mean square (RMS) were used as sEMG measures for fatigue and linear mixed effects model were applied. The MF was significantly decreased in fatigued condition, especially in the affected leg and in the tibialis anterior and peroneus longus muscle. The RMS did not change significantly in fatigued condition, while the OMNI fatigue score indicated patients felt really fatigued. No changes in sagittal kinematics of hip, knee and ankle were found using statistical non-parametric mapping. In conclusion, the current fatigue protocol seems promising in inducing fatigue in a population with CP with drop foot and it could be used to expand knowledge on muscle fatigue during walking in CP.

## Background

Cerebral palsy (CP) describes a group of permanent disorders of the development of movement and posture causing activity limitations, resulting from an injury in the developing central nervous system [1–3]. CP is the most common cause of motor disability in children (1.8–2.1 children per 1000 births) and most patients (±85%) experience spasticity [4–6]. During growth, the muscle length can fall short due to spasticity and bone deformities can develop [7].

**Funding:** There are no competing interests to declare. This study is part of a larger study entitled "Functional electrical stimulation of the ankle dorsiflexors during walking in children with unilateral spastic cerebral palsy: a randomized crossover intervention study" (ClinicalTrials.gov number NCT03440632) which is funded by HandicapNL (formerly Revalidatiefonds, project number R201605614, RJV), in collaboration with the Cornelia Stichting. No sponsor/funder played any role in study design, data collection and analysis, decision to publish and preparation of the manuscript.

**Competing interests:** No conflicts of interests to declare. The authors don't have financial relationships with other persons or organizations that might have inappropriately influenced our work presented in this manuscript.

As a consequence, abnormal gait patterns [8] with wasting of mechanical energy and co-activation and co-contractions of muscles develop. This makes walking less energy efficient in children with CP [9, 10], even in Gross Motor Function Classification System (GMFCS) level I [11, 12].

Children with CP often experience fatigue, restricting their activities and participation [13]. Several definitions of fatigue exist. Kluger et al. defines 'fatigue' as a subjective sensation, and 'fatigability' as objective, exercise-induced changes in performance [14]. Van Geel et al. defines fatigability as 'state fatigue': a form of fatigue that changes in response to tasks and circumstances, with a performance (objective) component and a perceived (subjective) component [15]. Physiologically, fatigability can be divided in two components: peripheral fatigue and central fatigue [16]. Peripheral fatigue is a loss in the force-generating capacity due to processes distal to the neuromuscular junction. Contributing factors include the muscles, glycogen storage, energy depletion and pathologies affecting the muscle functions (e.g. myasthenia gravis, Guillain-Barré syndrome) [14, 16]. Central fatigue is a progressive exercise-induced reduction in voluntary activation. Metabolic and structural lesions that disturb the normal process of activation in pathways between the basal ganglia, thalamus, limbic system and higher cortical centers contribute to the pathogenesis of central fatigue [17]. Children with CP are found to be less fatigue resistant than typically developing adolescents when testing submaximal contractions [18]. They have lower maximal muscle strength compared to typically developing children [19] and difficulties activating their muscles fully due to recruitment problems [20], as proved by lower activation levels during maximal voluntary contractions [21]. However, this means on the other side that children with CP potentially have a larger force reserve which can coincide with less peripheral fatigue [19, 20, 22]. As most activities in daily life are performed at submaximal strength, children with CP should probably be considered as being more fatigable [16]. Besides activation levels and force reserve, muscle morphology and structure can play a role in fatigue in CP. Several alterations in the latter have been described in CP, including reduced volume of muscles, shorter fascicle length and aberrant acetylcholine receptor localization [20]. Also, it has been reported that children with CP have more type I muscle fibers, which are slow but more fatigue resistant than type II fibers [23, 24].

Muscle fatigue can be assessed via surface electromyography (sEMG), measuring muscle activity using characteristics such as the root mean square (RMS) value to quantify the amplitude and the median frequency (MF) to quantify the spectral content of the sEMG signal [25]. These properties of the sEMG signal change during fatigue [26]. During muscle contractions metabolic products, such as lactic acid, are produced. Increased concentrations of lactic acid decreases the intracellular pH and consequently, the muscle fiber conduction velocity and MF decreases [26]. When muscles fatigue, the force-generating capacity decreases. Consequently, the neural drive increases to activate more motor units to maintain the required force level and this leads to an increase in RMS [16, 27]. Besides the decrease in conduction velocity, remaining activity of slow muscle fibers contributes to the decrease in MF and increase in RMS in fatigue [26].

Although fatigue is a frequently reported complaint in CP, little is known about measurements of fatigue during daily life activities such as walking [13]. Fatigue is mostly studied in (sub)maximal static contractions, which are less representative for daily life activities [16]. Knowledge about muscle fatigue during walking in CP is relevant because it could lead to new therapy insights to improve muscle endurance and finally activities in daily life. A few studies on fatigue during walking have been performed, reporting that more sEMG signs of muscle fatigue are present in the affected lower leg of children with CP during five minutes walking at self-selected speed compared to the non-affected leg or healthy children [16]. Parent et al. reported sEMG signs of muscle fatigue in the gluteus medius and rectus femoris muscle in

subjects with jump gait and crouch gait respectively during a six minute walking exercise at self-selected speed [28–30]. The aforementioned studies included patients with GMFCS level I-III. However, it is questionable how fatiguing it is to walk a few minutes at comfortable walking speed, especially for physically well-performing patients (GMFCS level I or II). We therefore developed a so called 'fatigue protocol' for children with CP with drop foot, to measure muscle fatigue after uphill and fast walking. These patients usually have a mild motor impairment (GMFCS level I) and can walk relatively well. Therefore, such a fatigue protocol might be more effective in inducing fatigue than walking at a comfortable speed at level ground. It should be taken into account that due to the uphill walking, it is possible that certain muscles will be relatively more fatigued, such as the hip- and knee flexors. It has been described that during uphill walking, the contribution to positive power decreases for the ankle and increases for the hip [31].

Besides analyzing changes in sEMG signals, it is important to analyze the impact of muscle fatigue on gait kinematics [28]. Parent et al. reported kinematic changes such as increased knee flexion in jump gait and crouch gait in children with CP in fatigued condition [28, 29]. No reports on kinematic changes due to fatigue in drop foot gait were found.

The aim of this study is to investigate whether the sEMG shows signs of fatigue (i.e. lower MF and higher RMS) after the fatigue protocol (compared to before) in children with CP with drop foot. The secondary aim is to investigate whether changes in sagittal plane kinematics of hip, knee and ankle occur after the fatigue protocol. We hypothesize that a lower MF and a higher RMS will be found after the fatigue protocol, especially in the hip flexors (M. rectus femoris) and the lower leg (M. gastrocnemius medialis, soleus, tibialis anterior and peroneus). Furthermore we hypothesize that no clinically relevant and statistically significant kinematic changes will occur in the sagittal plane for the ankle, knee and hip in this relatively well performing population.

## Methods

This study was part of the larger study entitled "Functional electrical stimulation of the ankle dorsiflexors during walking in children with unilateral spastic cerebral palsy: a randomized crossover intervention study". The Medical Ethical Committee of the MUMC approved the study (METC 172033/NL63250.068.17) and the trial was registered at ClinicalTrials.gov (NCT03440632). The full protocol was recently published [32].

### Participants

Eighteen patients diagnosed with unilateral spastic cerebral palsy, aged between 4 and 18 years, GMFCS level I or II and unilateral drop foot of central origin, currently treated with an ankle-foot-orthosis (AFO) or adapted shoes, were included between September 2018 –September 2021. All participants and/or caregivers (if the participant was < 16 years) signed an informed consent form before participating in the study.

### Procedure

For the main study all patients underwent two gait analyses with their conventional walking aid (mostly an AFO, otherwise adapted shoes), to support their drop foot during swing phase. In the current fatigue analysis, data from the first gait analysis with conventional walking aid was used, unless the second gait analysis had less missing data due to fallen sensors or poor signal to noise ratios. Gait analysis was performed at the 'Computer Assisted Rehabilitation Environment' (CAREN, MOTEK Medical, Amsterdam, Netherlands). The CAREN system consists of an dual-belt treadmill imbedded in a mobile platform with a force plate underneath

each belt, surrounded by a motion-capture system based on reflective markers, twelve infrared Bonita 3D-cameras (100 Hz, Vicon Nexus, Oxford, UK) and three 2D video cameras. A 3D virtual reality environment is projected on an 180˚ cylindrical screen. The Human Body Model (HBM) II lower limb kinematic model with trunk markers (26 markers [33]) was used. Furthermore, a 16 channel surface electromyography system (Delsys Trigno, see appendix S1 in S1 Text for technical details) was used to measure muscle activity in both legs: 1. M. rectus femoris; 2. M. vastus lateralis; 3. M. semitendinosus; 4. M. biceps femoris; 5. M. gastrocnemius medialis; 6. M. soleus; 7. M. tibialis anterior and 8. M. peroneus longus. sEMG sensors are placed according to the SENIAM guidelines [34]. The position of the sensors was verified by visually checking the raw sEMG signal during individual muscle contractions. A resting state sEMG measurement was performed while the patient was in supine, relaxed position. All hardware was integrated in D-flow software (MOTEK Medical, Amsterdam, Netherlands).

For this fatigue analysis, data from two conditions was used: 1) Comfortable walking speed with walking aid, here after called 'comfortable condition' 2) Comfortable walking speed after the fatigue protocol, with walking aid, here after called 'fatigued condition'. In every condition 250 steps (125 gait cycles) were measured. The comfortable walking speed with walking aid was determined by gradually increasing the speed of the treadmill with 0.01 m/s per second, starting at 0.5 m/s. The participant was asked to give a sign at the comfortable speed. This was repeated three times. The mean was considered the 'comfortable walking speed'.

## Fatigue protocol

The fatigue protocol was developed based on the Bruce protocol and the GMFCS treadmill protocols by Verschuren et al. [35, 36], to investigate the influence of walking-induced fatigue on the gait pattern. The patient was asked to give a score on the OMNI scale before the start of the fatigue protocol [37]. The OMNI scale ranges from 0 (totally not tired) to 10 (very, very tired). If the score was >2, a longer break for rest was implemented. During the fatigue protocol the speed and incline of the treadmill gradually increased and the patient was asked to walk as long as possible (Table 1 shows the details). If the patient indicated he could not walk further, despite motivational words, the protocol was stopped: the treadmill returned to a flat position. The final 125 gait cycles were immediately measured, at comfortable walking speed (in fatigued condition). At the moment the fatigue protocol was stopped, the patient was again asked to give a score on the children's OMNI scale of perceived exertion for his degree of fatigue.

## Data analysis

The raw data (C3d files) from the gait analyses were loaded into MATLAB (MATLAB R2019b, The Mathworks Inc., Natick, MA, USA). The raw sEMG signals were band-pass filtered at 10 to 250 Hz using a 4th order Butterworth filter to remove low frequency motion artefacts and high frequency noise [38]. Gait cycles were identified as heelstrike to heelstrike from the same

**Table 1. Details of the fatigue protocol.**

| Stage | Time (minutes) | % walking speed of the comfortable speed | Incline (˚) | Incline (%) |
|---|---|---|---|---|
| 1 | 3 (0–2:59) | 70% | 2 | 3.5 |
| 2 | 3 (3–5:59) | 85% | 4 | 7.0 |
| 3 | 3 (6–8:59) | 100% | 6 | 10.5 |
| 4 | 3 (9–11:59) | 115% | 8 | 14.1 |
| 5 | 3 (12–14:59) | 135% | 10 | 17.6 |
| 6 | 3 (15–17:59) | 140% | 12 | 21.3 |
| 7 | 3 (18–20:59) | 150% | 12 | 21.3 |

foot, based on the heel markers. Identified gait cycles were visually checked for consistency, where those typically shorter than 0.5 and longer than 1.5 seconds showed a deviation due to an incorrect additional or missing heelstrike. Gait cycles were then either manually corrected or removed to ensure that the included cycles were valid and consistent. Table S3 in the S1 Text shows the included number of gait cycles per patient per condition.

The MF is determined in Hertz as the median value of all valid gait cycles after individual Fourier transformations. Furthermore, sEMG was filtered with a root-mean-square window of 101ms, time normalized to 101 samples, and then averaged over all valid gait cycles. The RMS value in Volt is determined as the mean value of the active phase of a muscle based on the activation threshold. The activation threshold is set at the mean RMS value of the resting state sEMG signal plus two standard deviations, and hereafter called 'noise'. To ensure good quality of the EMG data, only muscles that reach an average signal-to-noise ratio (SNR) of at least 5 during walking are included in the analysis.

## Statistical analysis

Data distribution was checked using histograms and Q-Q plots. Patient characteristics and results of the fatigue protocol were described using mean and standard deviation. Changes in sEMG MF and RMS between the comfortable and fatigued condition were plotted in graphs with individual data and the mean with 95% confidence interval. Linear mixed effects models were applied for the MF and RMS separately, using the condition (comfortable or fatigued), the leg (affected or unaffected) and the muscle as fixed factors, an interaction term if applicable, a random intercept for subjects, and Bonferroni correction for multiple comparisons. These analyses were performed using IBM SPSS statistics version 24 (IBM Corp., Armonk, NY, USA).

Kinematics of the ankle, knee and hip in the sagittal plane were plotted using the group mean and standard deviations, based on individual means based on the valid cycles out of 125 gait cycles. Statistical (non)-parametric mapping (S(n)PM) was used to analyze possible differences between the comfortable and fatigued condition in kinematics over the whole gait cycle [39]. SPM is suitable for testing against critical values appropriate for smooth, spatially correlated time series, such as kinematics. This makes it possible to compare an entire waveform (0–100% of the gait cycle) instead of just specific timepoints. Technical details are provided elsewhere [40, 41]. SPM normality tests (spm1d.stats.normality.ttest_paired) were conducted for kinematics. Most of the kinematic curves were not normally distributed and therefore the statistical non-parametric mapping (SnPM) paired t-test (spm1d.stats.nonparam.ttest_paired) was used to compare the comfortable and fatigued condition. These analyses were done in MATLAB (*MATLAB R2019b*, *The Mathworks Inc.*, *Natick, MA, USA)*, using the SPM1D package (version 0.4, available at https://spm1d.org/index.html). For all SnPM tests the number of iterations was set at 1000 and statistically significant differences were not considered if they comprise ≤5% of the gait cycle, due to little clinical relevance [42]. A p-value of <0.05 was considered to be statistically significant for all analyses.

## Results

Eighteen children with CP were included (mean age 9.8 years, SD 4.1 years, n = 6 female), of whom sixteen were classified as GMFCS level I and two as GMFCS level II. The mean comfortable walking speed was 1.11 m/s (SD 0.22 m/s). The mean fulfilled time of the fatigue protocol was 614.7 seconds (SD 184.9 seconds) and the mean OMNI fatigue score after the fatigue protocol was 7.7 (SD 2.1). The individual values and patient characteristics can be found in Table 2.

**Table 2. Patient characteristics and results of the fatigue protocol.**

| Patient | GMFCS level | Sex | Conventional walking aid | Age in years | Comfortable walking speed (m/s) | Fulfilled time (min: sec) | Stage fatigue protocol | OMNI fatigue score (0–10) |
|---|---|---|---|---|---|---|---|---|
| 1 | I | M | D-AFO | 5 | 1.00 | 06:45 | 3 | 7 |
| 2 | I | M | D-AFO | 10 | 0.97 | 08:50 | 3 | 6 |
| 3* | I | M | D-AFO | 6 | 1.08 | 12:17 | 5 | 10 |
| 4 | I | M | Inlays | 6 | 1.13 | 06:00 | 3 | 4 |
| 5 | I | M | D-AFO | 8 | 1.20 | 13:03 | 5 | 10 |
| 6 | I | M | S-AFO | 17 | 1.48 | 6:20 | 3 | 7 |
| 7* | II | F | AFO | 16 | 0.79 | 08:14 | 3 | 8 |
| 8* | I | M | D-AFO | 4 | 0.65 | 12:00 | 5 | 9 |
| 9 | I | F | S-AFO | 9 | 1.15 | 12:30 | 5 | 10 |
| 10* | I | F | D-AFO | 13 | 1.28 | 09:50 | 4 | 7 |
| 11 | I | M | D-AFO | 14 | 1.23 | 04:30 | 2 | 4 |
| 12 | I | M | D-AFO | 6 | 1.08 | 11:22 | 4 | 8 |
| 13* | I | M | D-AFO | 13 | 1.10 | 14:20 | 5 | 8 |
| 14 | I | F | H-AFO | 10 | 1.28 | 10:39 | 4 | 9 |
| 15 | II | M | D-AFO | 15 | 1.42 | 08:28 | 3 | 5 |
| 16 | I | F | D-AFO | 11 | 1.20 | 12:37 | 5 | 9 |
| 17* | I | M | D-AFO | 5 | 0.90 | 11:10 | 4 | 10 |
| 18 | I | F | H-AFO | 8 | 0.78 | 15:53 | 6 | 6 |

Abbreviations used: GMFCS, Gross Motor Function Classification System; M, male; F, female; AFO, ankle-foot-orthosis; D, dynamic; H, hinged; S, solid. * = measurement 2 was used due to more missing data in measurement 1 (due to sensors that fell off or poor SNR).

A typical individual example of the raw sEMG signal is presented in Fig 1. For the affected leg, sEMG data of the lower leg is mostly not available due to the walking aid: in the case of an AFO, there is no space for the sEMG sensors on the m. gastrocnemius, m. soleus and m.

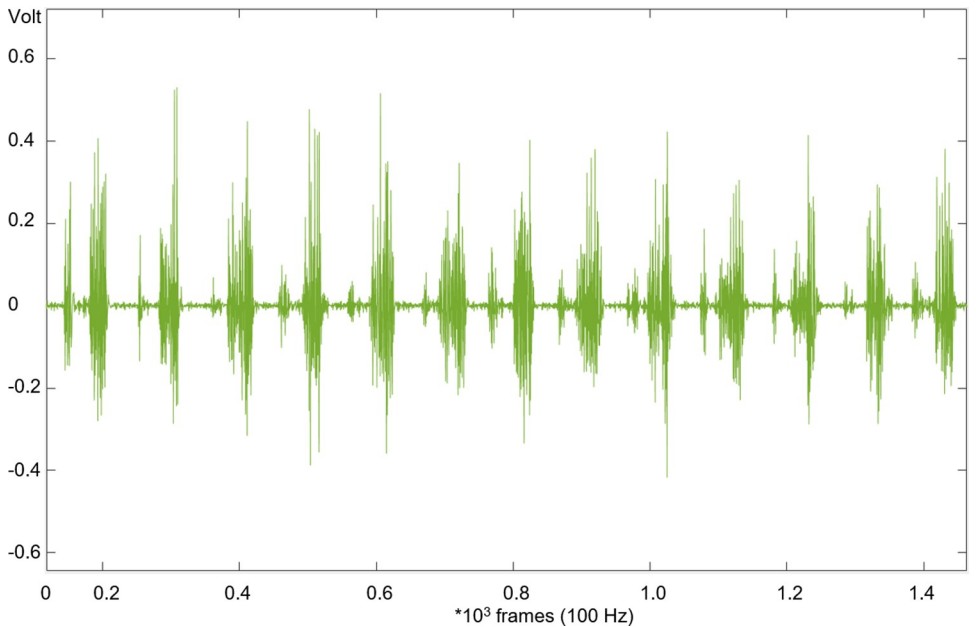

**Fig 1. Example of a raw sEMG signal of the gastrocnemius muscle.**

peroneus, and sometimes the tibialis anterior muscle. Also in two patients specific muscle signals had to be excluded, because sEMG sensors fell off during walking (5 sEMG sensors in total) e.g. due to sweating or the mechanical impact of foot-treadmill contact. Furthermore, in total 11 muscle signals of affected legs were excluded due to a low signal-to-noise ratio, and 33 muscle signals of unaffected legs, leaving sEMG data of 77 muscles of affected legs and 109 muscles of unaffected legs for analysis. Table 3 shows the numbers of available recordings per muscle.

Changes in sEMG MF and RMS between the comfortable and fatigued condition per muscle per patient are shown in Fig 2, both for the affected leg and unaffected leg. The MF was significantly predicted by the condition ($F_{(1, 329.10)}$ = 10.62, p = 0.001), by the muscle ($F_{(7, 333.39)}$ = 11.64, p<0.001), by the muscle x condition ($F_{(7, 329.10)}$ = 2.30, p = 0.027) and by the leg x condition ($F_{(1, 329.10)}$ = 4.62, p = 0.032) interaction terms. No statistically significant main effect for leg (affected/unaffected) was found ($F_{(1, 337.62)}$ = 0.35, p = 0.58). Pairwise comparisons showed that the MF significantly decreased by 10.4±3.2 Hz in the fatigued condition. The estimated marginal means per muscle per condition are displayed in Table 3. Pairwise comparisons for the muscle x condition interaction showed that the decrease in MF in the fatigued condition was stronger for the TibA (mean decrease 14.8±5.2 Hz, ($F_{(1, 298.34)}$ = 8.15, p = 0.005)) and the PerL (mean decrease 48.3±13.8 Hz, ($F_{(1, 298.34)}$ = 12.31, p = 0.001)) than for other muscles. Pairwise comparisons for the leg x condition interaction showed that the decrease in MF in the fatigued condition was stronger for the affected leg (mean decrease 16.8±6.1 Hz, ($F_{(1, 298.34)}$ = 7.61, p = 0.006) than for the unaffected leg.

The linear mixed-effects model for RMS didn't show any statistically significant effect of condition ($F_{(1, 325.66)}$ = 0.28, p = 0.60), muscle ($F_{(7, 335.79)}$ = 0.08, p = 1.0) or the leg ($F_{(1, 344.85)}$ = 0.15, p = 0.70) (Table 4).

Fig 3 shows the mean sagittal plane kinematics of the ankle, knee and hip for the comfortable condition and the fatigued condition, both for the affected and unaffected leg. SnPM

**Table 3. Estimated marginal means of the MF per muscle, based on the linear mixed-effects model.**

| Muscle | N | Condition | Mean (Hz) | Std error (Hz) | df | 95% confidence interval | | p-value |
|--------|---|-----------|-----------|----------------|-----|------------------------|--------------|---------|
| | | | | | | Lower bound | Upper bound | |
| RecF | 54 | Comfortable | 60.6 | 4.1 | 147.1 | 52.5 | 68.6 | 0.05 |
| | | Fatigued | 50.8 | | | 42.7 | 58.9 | |
| VasL | 52 | Comfortable | 58.0 | 4.1 | 151.7 | 49.8 | 66.2 | 0.58 |
| | | Fatigued | 55.2 | | | 47.0 | 63.4 | |
| SemT | 60 | Comfortable | 59.5 | 4.0 | 135.6 | 51.7 | 67.4 | 0.79 |
| | | Fatigued | 60.8 | | | 53.0 | 68.6 | |
| BicF | 62 | Comfortable | 69.9 | 3.9 | 126.5 | 62.3 | 77.6 | 0.13 |
| | | Fatigued | 62.9 | | | 55.3 | 70.6 | |
| GasM | 34 | Comfortable | 92.7 | 9.8 | 345.4 | 73.5 | 111.9 | 0.91 |
| | | Fatigued | 91.2 | | | 72.0 | 110.4 | |
| Sole | 32 | Comfortable | 85.4 | 9.8 | 345.5 | 66.2 | 104.6 | 0.95 |
| | | Fatigued | 84.6 | | | 65.4 | 103.8 | |
| TibA | 56 | Comfortable | 84.5 | 4.1 | 146.8 | 76.4 | 92.6 | <0.01 |
| | | Fatigued | 70.0 | | | 61.9 | 78.0 | |
| PerL | 28 | Comfortable | 105.6 | 9.8 | 345.4 | 86.3 | 124.9 | <0.01 |
| | | Fatigued | 57.4 | | | 38.1 | 76.7 | |

Abbreviations used: MF: median frequency; RecF: rectus femoris; VasL: vastus lateralis; SemT: semitendinosus; BicF: biceps femoris; GasM: gastrocnemius; Sole: soleus; TibA: tibialis anterior; PerL: peroneus longus.

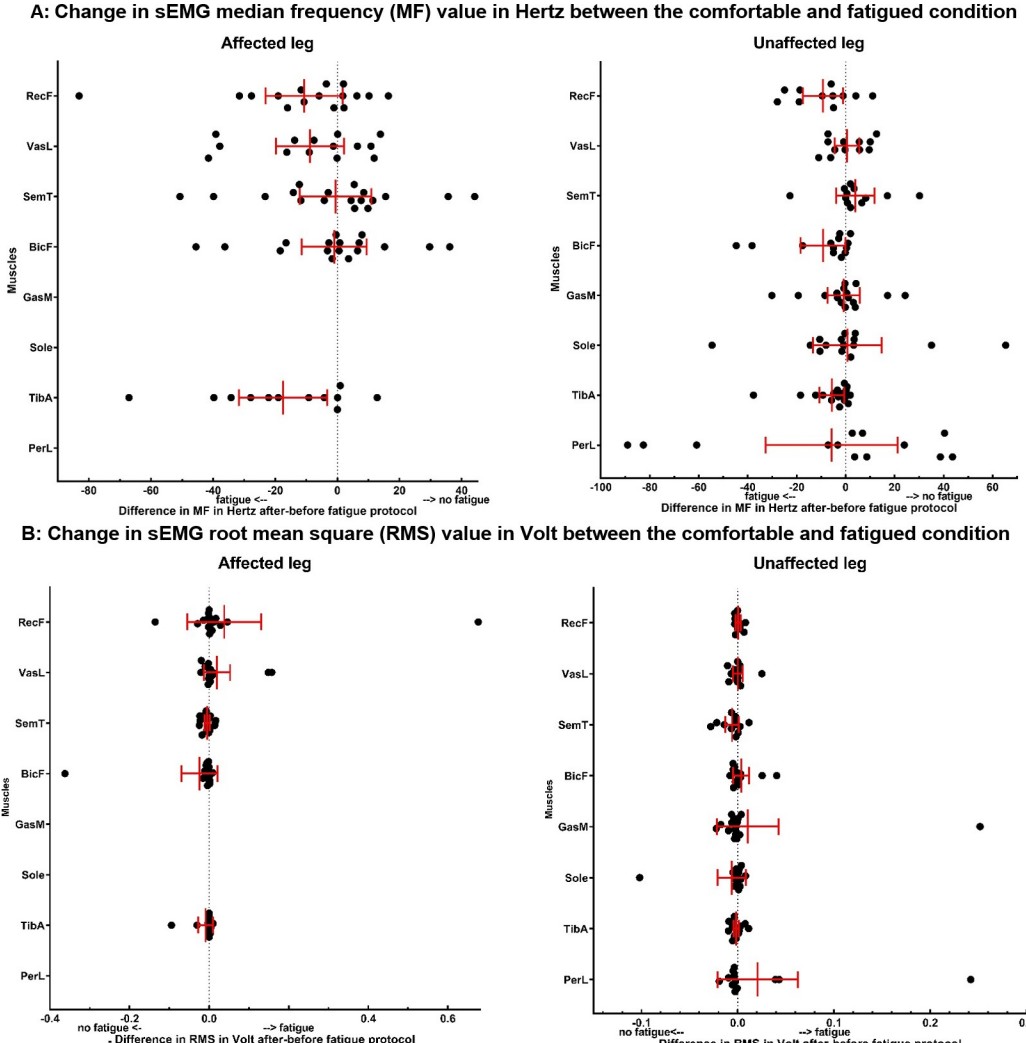

**Fig 2.** Changes in sEMG median frequency (MF, panel A) and root mean square (RMS, panel B) between the comfortable and fatigued condition; for the affected leg (graphs on the left) and unaffected leg (graphs on the right). The change is calculated as the difference between the value after the fatigue protocol minus the value before the fatigue protocol per muscle per patient and displayed on the x-axis. The y-axis shows the different muscles. *Abbreviations used: RecF: rectus femoris; VasL: vastus lateralis; SemT: semitendinosus; BicF: biceps femoris; GasM: gastrocnemius; Sole: soleus; TibA: tibialis anterior; PerL: peroneus longus.*

paired t-tests showed no statistically significant differences in kinematics between the comfortable and fatigued condition, except for the unaffected ankle (±1.3° more dorsiflexion during stance phase (40.0–47.5% of the gait cycle) in the fatigued condition) (See the appendix S2 in S1 Text for SnPM figures.

## Discussion

Based on the MF and the OMNI fatigue score, the current fatigue protocol seems promising in inducing fatigue in a population with CP and drop foot, while no changes RMS and in sagittal kinematics of hip, knee and ankle occurred.

The results rejected our first hypothesis because the RMS did not show a statistically significant increase in the fatigued condition, although the MF decreased significantly in the fatigued

**Table 4. Estimated marginal means of the RMS per muscle, based on the linear mixed-effects model.**

| Muscle | N | Condition | Mean (V) | Std error (V) | df | 95% confidence interval | |
|---|---|---|---|---|---|---|---|
| | | | | | | Lower bound | Upper bound |
| RecF | 54 | Comfortable | 0.022 | 0.010 | 294.185 | 0.002 | 0.041 |
| | | Fatigued | 0.045 | | | 0.026 | 0.064 |
| VasL | 52 | Comfortable | 0.025 | 0.010 | 299.360 | 0.005 | 0.044 |
| | | Fatigued | 0.035 | | | 0.015 | 0.054 |
| SemT | 60 | Comfortable | 0.032 | 0.009 | 290.844 | 0.013 | 0.050 |
| | | Fatigued | 0.027 | | | 0.008 | 0.046 |
| BicF | 62 | Comfortable | 0.036 | 0.009 | 284.211 | 0.018 | 0.054 |
| | | Fatigued | 0.026 | | 288.969 | 0.008 | 0.045 |
| GasM | 34 | Comfortable | 0.033 | 0.026 | 344.262 | -0.018 | 0.083 |
| | | Fatigued | 0.039 | | | -0.012 | 0.089 |
| Sole | 32 | Comfortable | 0.030 | 0.026 | 343.796 | -0.021 | 0.081 |
| | | Fatigued | 0.027 | | | -0.024 | 0.078 |
| TibA | 56 | Comfortable | 0.030 | 0.010 | 297.466 | 0.011 | 0.049 |
| | | Fatigued | 0.025 | | | 0.006 | 0.045 |
| PerL | 28 | Comfortable | 0.025 | 0.026 | 344.353 | -0.026 | 0.076 |
| | | Fatigued | 0.044 | | | -0.007 | 0.095 |

Abbreviations used: RMS: root mean square; RecF: rectus femoris; VasL: vastus lateralis; SemT: semitendinosus; BicF: biceps femoris; GasM: gastrocnemius; Sole: soleus; TibA: tibialis anterior; PerL: peroneus longus.

condition compared to the comfortable condition as expected. The decrease in MF in fatigued condition was stronger for the affected leg and for the m. tibialis anterior and m. peroneus longus. The rectus femoris muscle did not show statistically significant signs of fatigue in pairwise comparisons, rejecting our hypothesis that signs of fatigue would occur in the hip flexors after walking uphill.

Regarding kinematics, our hypothesis that no significant changes in sagittal plane kinematics of ankle, knee and hip would occur, was confirmed. The 1.3° increase in dorsiflexion in stance phase (40.0–47.5% of the gait cycle) in fatigued condition for the unaffected ankle is considered clinically irrelevant as it is smaller than the minimally detectable change [43, 44]. Thus, the fatigue protocol did not result in altered kinematics: the patients were still able to maintain their gait pattern (at the same walking speed). This result could be interpreted as an indication that the current study population is physically quite well-performing and that, despite the challenging fatigue protocol and sEMG signs of fatigue, the muscle fatigue did not limit their walking performance. It should be noted that the patients wore their walking aid (mostly an AFO) and this might have helped them to maintain a stable gait pattern. Despite the AFO, it was not impossible for kinematics to change, because most patients wore a dynamic AFO, which does allow some movement in the ankle joint. Besides, compensatory mechanisms such as changing step length or using circumduction could have been applied (leading to kinematic changes).

A decrease in MF and an increase in RMS are generally described to occur together in muscle fatigue. Cifrek et al. report that in the case of a decrease in MF, a decrease in RMS points to a decrease in muscle force, while an increase in RMS points to muscle fatigue [26]. However, fatigue and decrease in force are often related. In our study, the decrease in MF was apparently not accompanied by substantial decline in force, as the RMS values did not change due to the fatigue protocol, while the kinematics stayed the same. Moreover, factors such as perspiration can influence sEMG signals: it is reported that sweat decreases the RMS, but has less influence

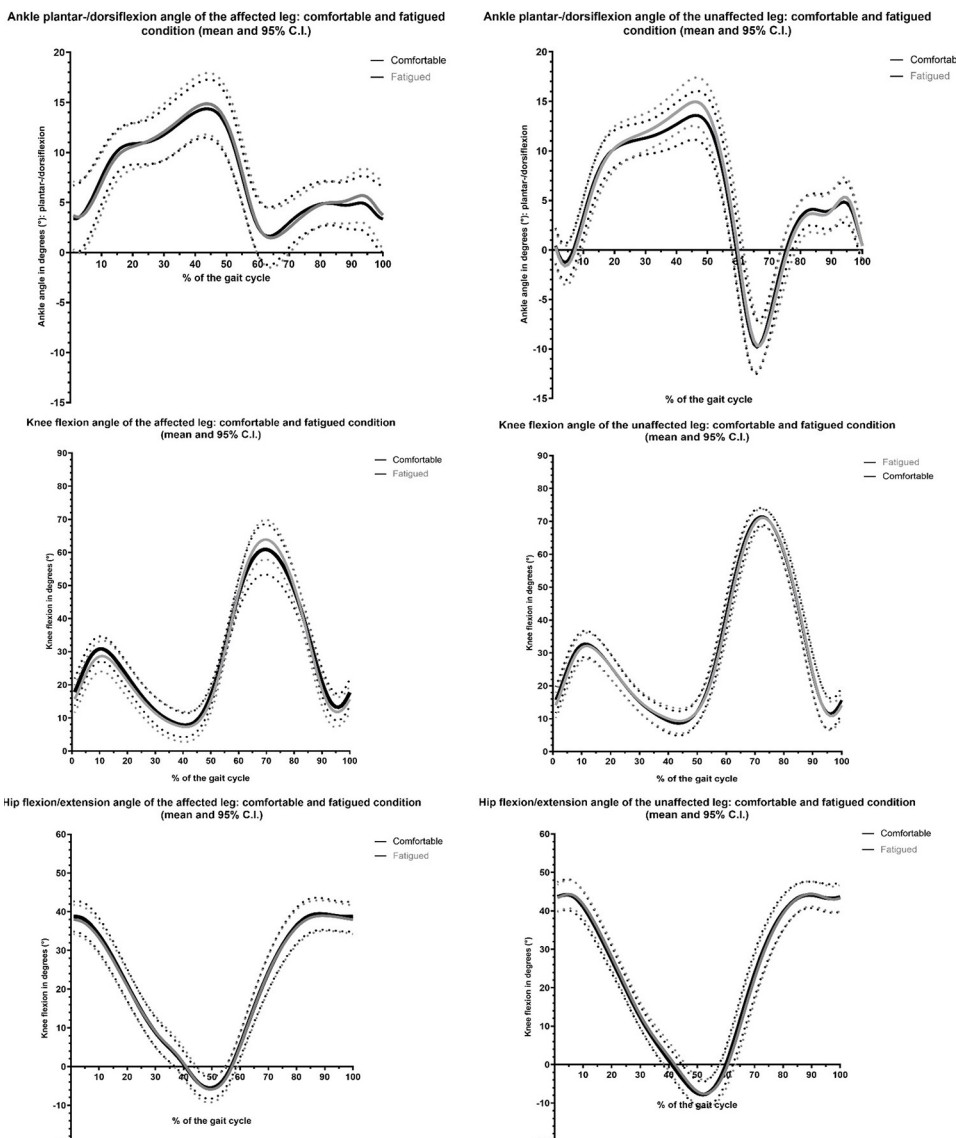

**Fig 3.** Mean sagittal plane kinematics of the ankle, knee and hip of the affected leg (graphs on the left) and unaffected leg (graphs on the right), for the comfortable (black line) and fatigued condition (grey line). N = 18.

on MF [45]. Some patients in our study were noticeably sweating during the fatigue protocol and this may have affected the RMS. Unfortunately we cannot correct for this. It is likely that both central and peripheral fatigue occur in children with CP. However, their contribution cannot be assessed based on the current sEMG measurements.

Our study has several limitations. First of all, due to the fact that the 3D gait analyses were performed with a walking aid, limited data of the muscles of the affected lower leg was available. It should be noted that conclusions regarding the ankle muscles were based on available data mostly from the unaffected leg. Even while data from the ankle muscles of the affected leg were mostly missing, the linear mixed effects model showed that the decrease in MF was stronger in the affected leg and (separately) in the TibA and PerL muscles. Based on previous findings [16], we expected that especially these muscles would show sEMG signs of fatigue in this population. Hence, the effects of this study may be underestimated and could have been

stronger if sEMG measurements of these muscles in the affected leg would have been included. Also, it would have been valuable if more muscles could have been included in sEMG measurement, especially the gluteus muscles, as hip extension contributes to positive power during uphill walking.

Secondly, we compared two conditions: walking in comfortable and fatigued condition at comfortable speed, without taking change over time into account. We did not assess whether fatigue was possibly already building up during the comfortable condition or declining in the fatigued condition (by comparing individual gait cycles). However, it was not the aim of this article to investigate how fast sEMG signs of fatigue occur or recover.

In the third place, it should be noted that patients walked uphill in order to induce fatigue. Walking uphill could induce fatigue in other muscles than walking on level ground. As stated before, the contribution to positive power increases for the hip and decreases for the ankle during uphill walking. Therefore, it could be expected that the rectus femoris (hip flexor) would be more fatigued in our study. This was however not the case. Apparently, even after uphill walking, muscle fatigue is still more pronounced in distal muscles in children with CP [46].

Finally, for the feasibility of the measurements, we didn't measure other objective parameters of the amount of physical strain, such as the heartrate or oxygen consumption to assess fatigue. Despite the lack of these objective parameters, we think that the fatigue protocol can be considered a 'maximal exercise', as are the Bruce and GMFCS treadmill protocols [35, 36]: the patient is forced to quit the task because of fatigue and exhausting [47]. This is especially the case when patients stop the protocol when they are really fatigued. The mean OMNI fatigue score of 7.7 reflects feeling really fatigued, however three patients only gave score 4 or 5 for fatigue. It seemed that one of these patients had some difficulties using the OMNI fatigue scale and the other two patients didn't seem to want to admit that they were really tired.

In conclusion, signs of fatigue are found after the fatigue protocol, namely a decrease in sEMG MF, especially in the affected leg and in the tibialis anterior and peroneus longus muscle, and a high OMNI fatigue score. Hence, the developed treadmill based fatigue protocol with increasing incline and speed, is effective in inducing fatigue in a population of children with CP (GMFCS level I-II). We think this fatigue protocol is suitable for future research, to expand the knowledge on muscle fatigue during walking in children with CP and its implications for gait and therapy. As expected, no kinematic changes were found in the fatigued condition for the ankle, knee and hip in the sagittal plane.

The current results are relevant for future research. First of all, after showing that the fatigue protocol is effective in inducing muscle fatigue, especially in the lower leg, this study offers a tailored protocol for children with CP to induce muscle fatigue. Second, this fatigue protocol makes it possible to evaluate changes in muscle fatigue after treatment or training programs. It could be investigated whether muscle strength and muscle spasticity, which are often treatment targets, are factors directly influencing muscle fatigue. Although we didn't find relevant kinematic changes, this study highlights the importance of taking kinematics into account, as they are directly influenced by muscle activity. Future research could focus on changes in kinematics during fatigue, with and without walking aid.

## Supporting information

**S1 Text. This file contains all the supporting tables and figures.**
(DOCX)

**S1 Data.**
(XLSX)

**S1 File.**

(ZIP)

**S2 File.**

(ZIP)

**S3 File.**

(ZIP)

## Acknowledgments

The authors are grateful to the participating children and their parents for their time and effort. The authors also thank Paul J.B. Willems for the technical implementation of the fatigue protocol at the CAREN. Furthermore, the authors want to thank M. Coenen, E. Claassen, R. Hovenier, D. Heijnen and E. Stoop for their help with the measurements.

## Author Contributions

**Conceptualization:** R. J. Vermeulen, K. Meijer.

**Data curation:** I. Moll, J. M. N. Essers, Y. J. M. Janssen-Potten.

**Formal analysis:** I. Moll, J. M. N. Essers.

**Investigation:** I. Moll, R. G. J. Marcellis, R. H. J. Senden, K. Meijer.

**Methodology:** Y. J. M. Janssen-Potten, K. Meijer.

**Supervision:** K. Meijer.

**Writing – original draft:** I. Moll.

**Writing – review & editing:** J. M. N. Essers, R. G. J. Marcellis, R. H. J. Senden, Y. J. M. Janssen-Potten, R. J. Vermeulen, K. Meijer.

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
