## [Decision Letter · Decision Letter 0]

9 Sep 2022

PONE-D-22-21864Lower limb muscle fatigue after uphill walking in children with unilateral spastic cerebral palsyPLOS ONE

Dear Dr. Moll,

Thank you for submitting your manuscript to PLOS ONE. After careful consideration, we feel that it has merit but does not fully meet PLOS ONE’s publication criteria as it currently stands. Therefore, we invite you to submit a revised version of the manuscript that addresses the points raised during the review process.

We look forward to receiving your revised manuscript.

Kind regards,

Xin Ye, Ph.D.

Academic Editor

PLOS ONE

Journal Requirements:

2. In the ethics statement in the Methods and online submission information, please ensure that you have specified (1) whether consent was informed and (2) what type you obtained (for instance, written or verbal, and if verbal, how it was documented and witnessed). If your study included minors, state whether you obtained consent from parents or guardians. If the need for consent was waived by the ethics committee, please include this information.

3. Please remove your figures from within your manuscript file, leaving only the individual TIFF/EPS image files, uploaded separately.  These will be automatically included in the reviewers’ PDF.

Reviewers' comments:

Reviewer's Responses to Questions

**Comments to the Author**

1. Is the manuscript technically sound, and do the data support the conclusions?

Reviewer #1: Yes

Reviewer #2: Partly

2. Has the statistical analysis been performed appropriately and rigorously? 

Reviewer #1: Yes

Reviewer #2: I Don't Know

3. Have the authors made all data underlying the findings in their manuscript fully available?

Reviewer #1: Yes

Reviewer #2: No

4. Is the manuscript presented in an intelligible fashion and written in standard English?

Reviewer #1: Yes

Reviewer #2: Yes

5. Review Comments to the Author

Reviewer #1: Thank you for the opportunity to review such a thoughtful experiment. The authors use well established protocols and methods to investigate a useful research question- the relationship among sEMG, perceived exertion, and gait kinematics. The authors encounter usual challenges with use of AFOs with gait analysis and application of sEMG and are able to still produce useful data.

The authors are encouraged to re examine the introduction, there are many statements about the muscles and fatigability of individuals with CP that are not referenced. There are also issues with changing tenses within the body of text of the methods. There may also be some literature that could be cited on energy expenditure and GMFCS I unilateral to provide context about the participants chosen for the fatigue protocol.

The authors miss an opportunity to provide a concise statement about their findings in the initial paragraph of the discussion. I like the way it is summarized in the abstract it is much clearer and powerful than the way it is described in the discussion.

Given all the differences in muscle morphology and physiology and fiber types how might you explain the findings? Please expand a little more what you think these findings tell us about central vs peripheral fatigue? Because of the OMNI scores do you think central fatigue is a bigger factor than peripheral?

Reviewer #2: This study presents results of original research that do not appear to have been published elsewhere. This is an interesting fatigue protocol for children with mild cerebral palsy. While the experiments appear to have been performed with high technical standards, additional detail is needed and some concerns for missing data need to be addressed before the conclusions can be justified.

Of greatest concern is the fact that EMG data for the ankle was completely missing for 3 out of 4 ankle muscles for the affected leg in all participants, yet conclusions were made regarding significant changes in frequency for two of the ankle muscles. It must be assumed that these decisions were determined with less than half the data available.

While there were 18 participants, all but one child wore an AFO during the assessment so at the most only one child would have data from both ankles. Therefore, no comparisons can be made between affected and unaffected ankles and any conclusions about the ankle responses is incomplete and represents only the unaffected leg.

The actual amount of data used for each participant is not clear. Data were collected for 250 steps before and after fatigue, however, only “valid steps” were included. More information is needed regarding how many valid steps were found for each participant under each condition. Gait cycles that were shorter than 0.5 or longer than 1.5 seconds were removed. Additional clarification and justification for this decision could be helpful to the reader.

The conclusions are presented in an appropriate fashion however, there are many unanswered questions that need to be resolved and further justification is needed to support the conclusions.

It is unclear why the authors expected to see fatigue related changes in kinematics when AFOs were worn for data collection. The AFO will dictate kinematics for the affected leg to a great extent and could also have interfered with muscle activity thus reducing potential for fatigue. This decision requires further explanation and justification.

Also, it is unclear why hip extensor or abductor emg data were not included in this data set since they should be activated and fatigued by uphill walking.

The article is presented in an intelligible fashion and is written in standard English, however, the authors should use people first language. Throughout the text they refer to children with cerebral palsy as ‘CP patients’.

Ethical standards appear to have been met with informed consent and IRB approval.

There are some problems with data availability. The supplementary data provided does not represent full disclosure of the data. Only a single, normalized mean was provided as kinematic data for each participant for each condition. Presumably hundreds of steps were included in creating that mean, however no individual step data is provided and it is unknown how many steps were considered valid for each participant.

EMG data were not accessible by this reviewer in the .sav file extension format. This was unreadable in wordpad or textedit. Is it possible to make the supplementary data more easily accessible to a wide variety of readers?

6. PLOS authors have the option to publish the peer review history of their article (what does this mean?). If published, this will include your full peer review and any attached files.

Reviewer #1: **Yes: **Mary E Gannotti

Reviewer #2: No

---

## [Author Response · Author response to Decision Letter 0]

17 Oct 2022

Reviewer #1: Thank you for the opportunity to review such a thoughtful experiment. The authors use well established protocols and methods to investigate an useful research question- the relationship among sEMG, perceived exertion, and gait kinematics. The authors encounter usual challenges with use of AFOs with gait analysis and application of sEMG and are able to still produce useful data.

The authors are encouraged to re-examine the introduction, there are many statements about the muscles and fatigability of individuals with CP that are not referenced. 

Thank you for your comments. Based on this remark, some references were added in the Background section (in red in the manuscript file):

GAGE, J. R., SCHWARTZ, M. H., KOOP, S. E. & NOVACHECK, T. F. 2009. Chapter 2.4 Consequences Of Brain Injury On Musculoskeletal Development. The Identification and Treatment of Gait Problems in Cerebral Palsy. Mac Keith Press.

EKEN, M. M., BRÆNDVIK, S. M., BARDAL, E. M., HOUDIJK, H., DALLMEIJER, A. J. & ROELEVELD, K. 2019. Lower limb muscle fatigue during walking in children with cerebral palsy.

EKEN, M. M., HOUDIJK, H., DOORENBOSCH, C. A., KIEZEBRINK, F. E., VAN BENNEKOM, C. A., HARLAAR, J. & DALLMEIJER, A. J. 2016. Relations between muscle endurance and subjectively reported fatigue, walking capacity, and participation in mildly affected adolescents with cerebral palsy. Developmental Medicine & Child Neurology, 58, 814-821.

There are also issues with changing tenses within the body of text of the methods. 

The tense has been changed to past tense for the whole section. 

There may also be some literature that could be cited on energy expenditure and GMFCS I unilateral to provide context about the participants chosen for the fatigue protocol.

It is true that some more specific attention could be put on GMFCS level I, although the used references give information on GMFCS level I. It was now emphasized that the energy expenditure is also higher in GMFCS I compared to typically developing children: ‘This makes walking less energy efficient in CP patients (Rose et al., 1990, Ratel et al., 2019), even in Gross Motor Function Classification System (GMFCS) level I (Dallmeijer and Brehm, 2011, Johnston et al., 2004)’

The authors miss an opportunity to provide a concise statement about their findings in the initial paragraph of the discussion. I like the way it is summarized in the abstract it is much clearer and powerful than the way it is described in the discussion.

Thank you for the suggestion. The first sentence of the discussion has been changed: Based on the MF and the OMNI fatigue score, the current fatigue protocol seems promising in inducing fatigue in a CP population with drop foot, while no changes in RMS and sagittal kinematics of hip, knee and ankle occurred.

Given all the differences in muscle morphology and physiology and fiber types how might you explain the findings? Please expand a little more what you think these findings tell us about central vs peripheral fatigue? Because of the OMNI scores do you think central fatigue is a bigger factor than peripheral?

It could be stated that a decrease in MF reflects peripheral fatigue (related to accumulation of metabolic products in the muscle) and that an increase in RMS reflects central fatigue (increase in neural drive). The fact that the kinematics stayed stable in our study, could suggest that the force generating capacity (peripheral fatigue) remained sufficient, although the MF decreased. The OMNI score reflects subjective fatigue, which can both be central and peripheral. However, as stated, the contribution of central and peripheral fatigue cannot be assessed based on our measurements, because factors such as perspiration, which can decrease the RMS, should also be taken in to account. We think central and peripheral fatigue should be considered as occurring together. 

Reviewer #2: This study presents results of original research that do not appear to have been published elsewhere. This is an interesting fatigue protocol for children with mild cerebral palsy. While the experiments appear to have been performed with high technical standards, additional detail is needed and some concerns for missing data need to be addressed before the conclusions can be justified. Of greatest concern is the fact that EMG data for the ankle was completely missing for 3 out of 4 ankle muscles for the affected leg in all participants, yet conclusions were made regarding significant changes in frequency for two of the ankle muscles. It must be assumed that these decisions were determined with less than half the data available.

It is correct that most data on ankle muscles of the affected leg was missing due to AFOs, and that results on ankle muscles were therefore based on less data. This was also stated in the discussion (Limitations section): ‘Our study has several limitations. First of all, due to the fact that the 3D gait analyses were performed with a walking aid, limited data of the muscles of the affected lower leg was available. Based on previous findings (Eken et al., 2019), we expected that especially these muscles would show sEMG signs of fatigue in this population. Hence, the effects of this study may be underestimated and could have been stronger if sEMG measurements of these muscles would have been included.’ 

To be more clear about the missing data, the text has been adjusted: 

‘Our study has several limitations. First of all, due to the fact that the 3D gait analyses were performed with a walking aid, limited data of the muscles of the affected lower leg was available. It should be noted that conclusions regarding the ankle muscles were based on available data mostly from the unaffected leg. Even while data from the ankle muscles of the affected leg were mostly missing, the linear mixed effects model showed that the decrease in MF was stronger in the affected leg and (separately) in the TibA and PerL muscles. Based on previous findings (Eken et al., 2019), we expected that especially these muscles would show sEMG signs of fatigue in this population. Hence, the effects of this study may be underestimated and could have been stronger if sEMG measurements of these muscles in the affected leg would have been included.’

While there were 18 participants, all but one child wore an AFO during the assessment so at the most only one child would have data from both ankles. Therefore, no comparisons can be made between affected and unaffected ankles and any conclusions about the ankle responses is incomplete and represents only the unaffected leg.

This is correct again, although data on the tibialis anterior muscle (one of the four ankle muscles) was available for the affected leg in most cases. We refer to the added text in the answer above. We think a comparison between the affected leg and unaffected leg can be made, because for five muscles data from both leg were more or less completely available. Also, the linear mixed effects model does not compare the ankle muscles in the affected and unaffected leg, but compares the legs in total.

The actual amount of data used for each participant is not clear. Data were collected for 250 steps before and after fatigue, however, only “valid steps” were included. More information is needed regarding how many valid steps were found for each participant under each condition. 

Thank you for your question. A table with the number of included valid gait cycles per patient per condition was added as Appendix S3. 

S3: Number of valid gait cycles per patient per condition 

Patient number F-Patient number** Affected side Condition Number of valid gait cycles

1 2 Right Comfortable 127

1 2 Right Fatigued 62

2 3 Left Comfortable 127

2 3 Left Fatigued 149

3* 4* Left Comfortable 129

3* 4* Left Fatigued 128

4 5 Left Comfortable 127

4 5 Left Fatigued 128

5 6 Left Comfortable 129

5 6 Left Fatigued 130

6 8 Right Comfortable 128

6 8 Right Fatigued 121

7* 9* Right Comfortable 127

7* 9* Right Fatigued 121

8* 10* Left Comfortable 136

8* 10* Left Fatigued 131

9 12 Right Comfortable 138

9 12 Right Fatigued 129

10 13 Left Comfortable 133

10 13 Left Fatigued 131

11 14 Right Comfortable 146

11 14 Right Fatigued 159

12 18 Left Comfortable 130

12 18 Left Fatigued 128

13* 19* Left Comfortable 128

13* 19* Left Fatigued 144

14 20 Right Comfortable 128

14 20 Right Fatigued 128

15 21 Right Comfortable 133

15 21 Right Fatigued 130

16 23 Right Comfortable 138

16 23 Right Fatigued 129

17* 24* Left Comfortable 129

17* 24* Left Fatigued 129

18 25 Left Comfortable 136

18 25 Left Fatigued 140

* = measurement 2 was used due to more missing data in measurement 1 (due to sensors that fell off or poor SNR). ** = the original patient number, also used in the individual kinematic Excel files (Supplementary data).

Gait cycles that were shorter than 0.5 or longer than 1.5 seconds were removed. Additional clarification and justification for this decision could be helpful to the reader.

Thank you for your question. The text in the manuscript has been elaborated: 

Original: ‘Gait cycles are identified as heelstrike to heelstrike from the same foot, based on the heel markers. Identified gait cycles shorter than 0.5 or longer than 1.5 seconds were removed to ensure that the included steps were valid and consistent.’

New version: ‘Gait cycles were identified as heelstrike to heelstrike from the same foot, based on the heel markers. Identified gait cycles were visually checked for consistency, where those typically shorter than 0.5 and longer than 1.5 seconds showed a deviation due to an incorrect additional or missing heelstrike. Gait cycles were then either manually corrected or removed to ensure that the included cycles were valid and consistent.’

The conclusions are presented in an appropriate fashion however, there are many unanswered questions that need to be resolved and further justification is needed to support the conclusions.

It is unclear why the authors expected to see fatigue related changes in kinematics when AFOs were worn for data collection. The AFO will dictate kinematics for the affected leg to a great extent and could also have interfered with muscle activity thus reducing potential for fatigue. This decision requires further explanation and justification.

First of all, the hypothesis was that no significant changes in the sagittal plane kinematics for ankle, knee and hip would occur. We feel that it is incomplete however to analyze changes in sEMG signals without taking kinematics in consideration. It is true that AFOs reduce the opportunity of changes in kinematics and reduce the muscle activity of some muscles. However, our data still shows muscle activity of the tibialis anterior muscle in the affected leg while wearing an AFO. Most patients wore a dynamic AFO, which does allow some movement in the ankle joint. Besides, patients could have applied compensatory mechanisms in walking, e.g. changing step length or using circumduction, which would have led to changes in at least one of the three joints. 

Also, it is unclear why hip extensor or abductor emg data were not included in this data set since they should be activated and fatigued by uphill walking.

It is true that it would have been valuable to include sEMG of the gluteus muscles. We did collect some information on hip extensors as the semitendinosus and biceps femoris muscles also extend the hip. Unfortunately, we could not use more than 16 sEMG sensors due to availability of equipment. A sentence has been added in the limitation section: ‘Also, it would have been valuable if more muscles could have been included in sEMG measurement, especially the gluteus muscles, as hip extension contributes to positive power during uphill walking.’

The article is presented in an intelligible fashion and is written in standard English, however, the authors should use people first language. Throughout the text they refer to children with cerebral palsy as ‘CP patients’. Ethical standards appear to have been met with informed consent and IRB approval.

You are right about people first language, thank you for the comment. ‘CP patients’ has been replaced by ‘children with CP’ throughout the text. 

There are some problems with data availability. The supplementary data provided does not represent full disclosure of the data. Only a single, normalized mean was provided as kinematic data for each participant for each condition. Presumably hundreds of steps were included in creating that mean, however no individual step data is provided and it is unknown how many steps were considered valid for each participant.

A table with the number of gait cycles per patient per condition has been added in Appendix S3 (see above). Also the individual kinematic data have been uploaded as Supplementary data (Zip files with Excel files per patient per condition). 

EMG data were not accessible by this reviewer in the .sav file extension format. This was unreadable in wordpad or textedit. Is it possible to make the supplementary data more easily accessible to a wide variety of readers?

Sorry for the inconvenience. The EMG data was first provided in SPSS files, this has been changed to xlsx files (Excel).

---

## [Decision Letter · Decision Letter 1]

22 Nov 2022

Lower limb muscle fatigue after uphill walking in children with unilateral spastic cerebral palsy

PONE-D-22-21864R1

Dear Dr. Moll,

We’re pleased to inform you that your manuscript has been judged scientifically suitable for publication and will be formally accepted for publication once it meets all outstanding technical requirements.

Kind regards,

Xin Ye, Ph.D.

Academic Editor

PLOS ONE

Additional Editor Comments (optional):

Reviewers' comments:

Reviewer's Responses to Questions

**Comments to the Author**

1. If the authors have adequately addressed your comments raised in a previous round of review and you feel that this manuscript is now acceptable for publication, you may indicate that here to bypass the “Comments to the Author” section, enter your conflict of interest statement in the “Confidential to Editor” section, and submit your "Accept" recommendation.

Reviewer #1: All comments have been addressed

Reviewer #2: All comments have been addressed

2. Is the manuscript technically sound, and do the data support the conclusions?

Reviewer #1: Yes

Reviewer #2: Yes

3. Has the statistical analysis been performed appropriately and rigorously? 

Reviewer #1: Yes

Reviewer #2: Yes

4. Have the authors made all data underlying the findings in their manuscript fully available?

Reviewer #1: Yes

Reviewer #2: Yes

5. Is the manuscript presented in an intelligible fashion and written in standard English?

Reviewer #1: Yes

Reviewer #2: Yes

6. Review Comments to the Author

Reviewer #1: Congratulations on a well presented study. The authors have addressed my concerns and I believe the paper adds to our understanding of fatigue among children with cerebral palsy.

Reviewer #2: thank you for attention to details and responding to all reviewer questions in a thoughtful and transparent manner

7. PLOS authors have the option to publish the peer review history of their article (what does this mean?). If published, this will include your full peer review and any attached files.

Reviewer #1: **Yes: **Mary E Gannotti

Reviewer #2: No

---

## [Editor Report · Acceptance letter]

28 Nov 2022

PONE-D-22-21864R1 

Lower limb muscle fatigue after uphill walking in children with unilateral spastic cerebral palsy 

Dear Dr. Moll:

I'm pleased to inform you that your manuscript has been deemed suitable for publication in PLOS ONE. Congratulations! Your manuscript is now with our production department. 

Kind regards, 

on behalf of

Dr. Xin Ye 

Academic Editor

PLOS ONE